# Does diversity beget diversity in microbiomes?

**Naïma Madi[1], Michiel Vos[2], Carmen Lia Murall[1], Pierre Legendre[1], B Jesse Shapiro[1,3,4]\***

[1]Département de sciences biologiques, Université de Montréal, Montreal, Canada; [2]European Centre for Environment and Human Health, University of Exeter, Penryn, United Kingdom; [3]Department of Microbiology and Immunology, McGill University, Montreal, Canada; [4]McGill Genome Centre, McGill University, Montreal, Canada

**Abstract** Microbes are embedded in complex communities where they engage in a wide array of intra- and inter-specific interactions. The extent to which these interactions drive or impede microbiome diversity is not well understood. Historically, two contrasting hypotheses have been suggested to explain how species interactions could influence diversity. 'Ecological Controls' (EC) predicts a negative relationship, where the evolution or migration of novel types is constrained as niches become filled. In contrast, 'Diversity Begets Diversity' (DBD) predicts a positive relationship, with existing diversity promoting the accumulation of further diversity via niche construction and other interactions. Using high-throughput amplicon sequencing data from the Earth Microbiome Project, we provide evidence that DBD is strongest in low-diversity biomes, but weaker in more diverse biomes, consistent with biotic interactions initially favouring the accumulation of diversity (as predicted by DBD). However, as niches become increasingly filled, diversity hits a plateau (as predicted by EC).

**\*For correspondence:**
jesse.shapiro@mcgill.ca

**Competing interests:** The authors declare that no competing interests exist.

## Introduction

The majority of the genetic diversity on Earth is encoded by microbes (*Hug et al., 2016*; *Lapierre and Gogarten, 2009*; *Tara Oceans coordinators et al., 2015*) and the functioning of all Earth's ecosystems is reliant on diverse microbial communities (*Falkowski et al., 2008*). High-throughput 16S rRNA gene amplicon sequencing studies continue to yield unprecedented insight into the taxonomic richness of microbiomes (e.g. *Louca et al., 2019*; *Sogin et al., 2006*), and abiotic drivers of community composition (e.g. pH; *Lauber et al., 2009*; *Power et al., 2018*) are increasingly characterized. Although it is known that biotic (microbe-microbe) interactions can also be important in determining community composition (*Needham and Fuhrman, 2016*), comparatively little is known about how such interactions, either positive (e.g. cross-feeding; *Seth and Taga, 2014*) or negative (e.g. toxin-mediated interference competition; *Czárán et al., 2002*; *Hibbing et al., 2010*), shape microbiome diversity as a whole.

The dearth of studies exploring how microbial interactions could influence diversity stands in marked contrast to a long research tradition on biotic controls of plant and animal diversity (*Elton, 1946*; *Gause, 2003*). In an early study of 49 animal (vertebrate and invertebrate) community samples, *Elton, 1946* plotted the number of species versus the number of genera and observed a ~ 1:1 ratio in each individual sample, but a ~ 4:1 ratio when all samples were pooled. He took this observation as evidence for competitive exclusion preventing related species, more likely to overlap in niche space, to co-exist. This concept, more recently referred to as niche filling or Ecological Controls (EC) (*Schluter and Pennell, 2017*), predicts speciation (or, more generally, diversification) rates to decrease with increasing standing species diversity because less niche space is available (*Rabosky and Hurlbert, 2015*). In contrast, the Diversity Begets Diversity (DBD) model predicts that

when species interactions create novel niches, standing biodiversity favours further diversification (*Calcagno et al., 2017*; *Whittaker, 1972*). For example, niche construction (i.e. the physical, chemical or biological alteration of the environment) could influence the evolution of the species constructing the niche, as well as that of co-occurring species (*Laland et al., 1999*; *San Roman and Wagner, 2018*). An alternative to either EC or DBD is The Neutral Theory of Biodiversity and Biogeography, in which all species are functionally equivalent and communities assemble via random sampling (*Hubbell, 2001*). Neutral Theory serves as a null hypothesis of community assembly in macrobes (*Azaele et al., 2016*; *Gotelli and McGill, 2006*), and more recently in microbiome research (*Harris et al., 2017*; *Li and Ma, 2016*).

Empirical evidence for the action of EC vs. DBD in natural plant and animal communities has been mixed (*Calcagno et al., 2017*; *Emerson and Kolm, 2005*; *Palmer and Maurer, 1997*; *Price et al., 2014*; *Rabosky et al., 2018*). Laboratory evolution experiments tracking the diversification of a focal bacterial lineage in communities of varying complexity have also yielded contradictory results, with support for EC, DBD, or intermediate scenarios (*Brockhurst et al., 2007*; *Meyer and Kassen, 2007*). For example, diversification of a focal *Pseudomonas* clone was favoured by increasing community diversity in the range of 0–20 other strains or species within the same genus (*Calcagno et al., 2017*; *Jousset et al., 2016*) but diversification was inhibited in highly diverse communities (e.g. hundreds or thousands of species in compost; *Gómez and Buckling, 2013*). These experiments are consistent with interspecific competition initially driving (*Bailey et al., 2013*), but eventually inhibiting diversification as niches are filled.

Most laboratory experiments are restricted to relatively short evolutionary time scales and include only a small number of taxa; it is therefore unclear if they can be generalized to natural communities consisting of many more taxa evolving and assembling over much longer periods, spanning more environmental change, greater evolutionary diversification, and frequent migration events. Although the absence of a substantial prokaryotic fossil record hinders deconvoluting speciation and extinction rates (*Louca and Pennell, 2020*; *Marshall, 2017*) *Louca et al., 2018* recently estimated that bacterial diversity has mostly increased over the past billion years, with speciation rates slightly exceeding extinction rates. However, because many free-living microbes have high migration rates ('everything is everywhere, but the environment selects' [*de Wit and Bouvier, 2006*]), we expect that the majority of diversity present within a typical microbiome sample is selected from a pool of migrants rather than having evolved in situ. As such, here we broadly define 'diversity begets diversity' (DBD) to include the combined effects of community assembly from a migrant pool ('ecological species sorting') and in situ evolutionary diversification (*Figure 1*).

To test whether patterns of diversity in natural communities conform to EC or DBD dynamics, we used 2000 microbiome samples from the Earth Microbiome Project (EMP), the largest available repository of biodiversity based on standardized sampling and sequencing protocols, with 16S rRNA gene amplicon sequence variants (ASVs) as the finest-grained taxonomic unit (*Thompson et al., 2017*). Following *Elton, 1946*, we use the equivalent of Species:Genus ratios, calculating a range of taxonomic diversity ratios (up to the Class:Phylum level) as proxies for diversity within a focal taxon, from shallow to deep evolutionary time. We then plot each ratio as a function of the number of non-focal taxa (Genera, Families, Orders, Classes, and Phyla, respectively) with which the focal taxon could interact. We refer to the slope of these plots as the 'diversity slope', with negative slopes supporting EC and positive slopes supporting DBD (*Figure 1*). As a null, we compare these slopes to the expectation under Neutral Theory. To avoid a trivially positive diversity slope due to variation in sequencing effort, all samples were rarefied to 5000 observations (counts of 16S rRNA gene sequences), as diversity estimates are highly sensitive to sampling effort (*Gotelli and Colwell, 2001*). As 16S evolves at a rate of roughly 1–2 substitutions per million years (*Kuo and Ochman, 2009b*), evolutionary diversification within individual EMP samples cannot be uncovered using this marker; rather our data represent mainly a record of community assembly.

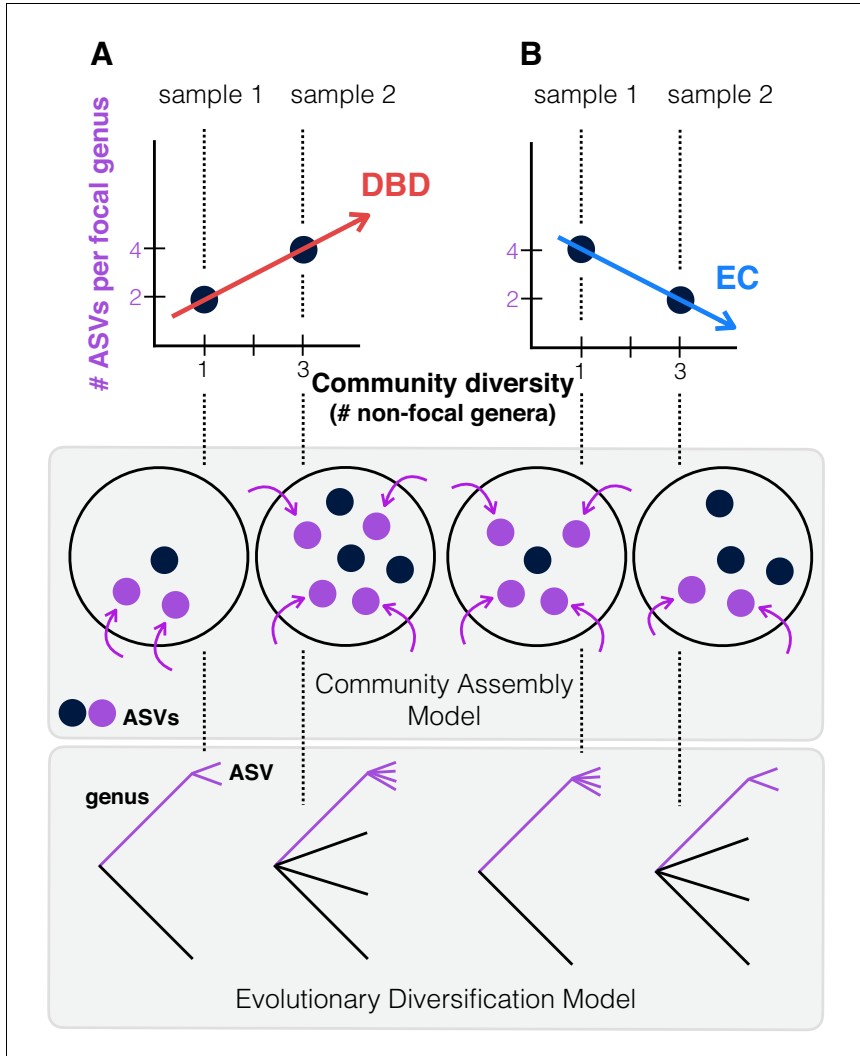

**Figure 1.** Contrasting the Diversity Begets Diversity (DBD) and Ecological Controls (EC) models. (**A**). In this hypothetical scenario, microbiome sample 1 contains one non-focal genus, and two amplicon sequence variants (ASVs) within the focal genus (point at x = 1, y = 2 in the plot). Sample 2 contains three non-focal genera, and four ASVs within the focal genus (point at x = 3, y = 4). Tracing a line through these points yields a positive diversity slope, supporting the DBD model (red). (**B**) Alternatively, a negative slope would support the Ecological Controls (EC) model (blue line). In the middle panel, we consider a community assembly model to explain the hypothetical data of the top panel, in which standing diversity (black points) in a community selects (for or against) new types (referred to here as ASVs) which arrive via migration (purple points and arrows). In the bottom panel, we consider an evolutionary diversification model of a focal lineage (genus) into ASVs as a function of initial genus-level community diversity present at the time of diversification.

## Results

### Quantifying the DBD-EC continuum in prokaryote communities compared to neutral null models

We used generalized linear mixed models (GLMMs) to estimate the diversity slope at each taxonomic level in the EMP data, which revealed a tendency towards positive slopes with significant variation explained by the random effects of lineage, environment, and their interaction (*Table 1*, *Figure 2*, *Figure 2—figure supplement 1–16*, *Supplementary file 1* Section 1). All models reported here provide significantly better fits compared to models without the fixed effect of community diversity, and coefficients of determination ($R^2$) are higher with the inclusion of random effects,

**Table 1.** Effects of community diversity on focal lineage diversity across taxonomic ratios.

The GLMMs show a statistically significant positive effect of community diversity on focal lineage diversity. Each row reports the effect of community diversity (Div) on focal lineage diversity, as well as its standard error, Wald z-statistic for its effect size and the corresponding *P*-value (left section), or standard deviation on the slope for the significant random effects (right section). SE = standard error, Env = environment type, Lin = lineage type, Lab = Principal Investigator ID, Sample = EMP Sample ID. Interactions are denoted as '*'. n.s. = not significant (likelihood-ratio test). All models provide a significantly better fit than null models without fixed effects ($\Delta$AIC > 10 and p<0.05; *Supplementary file 2*).

| | Slope (fixed effects) | | | | Standard deviation on the slope (random effects) | | | | |
|---|---|---|---|---|---|---|---|---|---|
| | Div | SE | z | P | Env | Lin | Lin*Env | Env*Lab | Sample |
| ASV:Genus | 0.091 | 0.016 | 5.792 | 6.95e-09 | n.s. | 0.074 | 0.142 | 0.114 | 0.067 |
| Genus:Family | 0.047 | 0.008 | 5.911 | 3.41e-09 | n.s. | 0.071 | 0.07 | 0.039 | n.s. |
| Family:Order | 0.119 | 0.017 | 7.001 | 2.54e-12 | 0.023 | 0.094 | 0.092 | 0.106 | n.s. |
| Order:Class | 0.109 | 0.020 | 5.447 | 5.13e-08 | 0.05 | 0.141 | 0.078 | 0.051 | n.s. |
| Class:Phylum | 0.272 | 0.043 | 6.341 | 2.29e-10 | 0.119 | 0.174 | 0.119 | 0.114 | n.s. |

showing their importance (*Supplementary file 2*). Examples of how the diversity slope varies across lineages and environments are shown in *Figure 2* and *Figure 2—figure supplement 2–16*. To assess the significance of these slope estimates in light of potential sampling bias and data structure (*Gotelli and Colwell, 2001*; *Jarvinen, 1982*), we considered null models, all of which randomize the associations between ASVs within a sample, thus randomizing any true biotic interactions. Models 1 and 2 are based on draws from the zero-sum multinomial (ZSM) distribution, which arises from the standard Neutral Theory of Biodiversity (Materials and methods). Model 1, in which each microbiome sample is drawn from the same ZSM distribution, produces a significantly negative diversity slope (*Figure 2—figure supplement 17*; *Table 2*). Model 2, in which each environment draws from a separate distribution, is effectively a composite of Model 1 in which different environments, each with a negative slope, are 'stacked' to yield an overall positive slope (*Figure 2—figure supplement 17*). However, the Model 2 slope is not significant in a GLMM accounting for variation across environments (*Table 2*, *Supplementary file 3* Section 1.2). In the real EMP data, most individual environments tend towards a positive slope (*Figure 2—figure supplement 18*). The tendency towards positive diversity slopes in the EMP is therefore not straightforwardly explained by neutral processes.

To estimate the power to detect either DBD or EC, we specifically added each of these effects to data simulated under a null model. As expected, adding DBD reversed the negative slope and rendered it positive (*Table 2*; *Figure 2—figure supplement 17*, *Supplementary file 3* Section 2.1), suggesting reasonable power to detect DBD when truly present. In contrast, the addition of EC had little effect on the slope, suggesting low power to detect EC under some null models. Taken together, these modelling results suggest that positive diversity slopes observed in the EMP are more readily explained by DBD than by Neutral Theory, whereas negative slopes could be explained by EC, Neutral Theory, or some combination of the two.

Because taxonomic labels can be unavailable or inconsistent with phylogenetic relationships (*Parks et al., 2018*; *Vos, 2011*) we repeated the analyses using nucleotide sequence identity in the 16S rRNA gene instead of taxonomy, and again recovered generally positive diversity slopes (Materials and methods). As a final sensitivity analysis, we repeated the GLMMs using unrarefied community Shannon diversity instead of richness (Materials and methods) and obtained similar results, with generally positive diversity slopes that could in some cases be reversed depending on the lineage or environment (*Table 3*, *Supplementary file 1* Section 2). The Shannon diversity metric is robust to sampling effort, suggesting that the results are not biased by undersampling in diverse biomes. Even if undersampling could bias the diversity slope downward in more diverse samples, the effect is unlikely to be large at a rarefaction to 5000 sequences, and only to occur at the extremes of diversity (e.g. very many genera and high ASV:genus ratios) and not at higher taxonomic levels (e.g. Class:Phylum) (*Figure 2—figure supplement 19*).

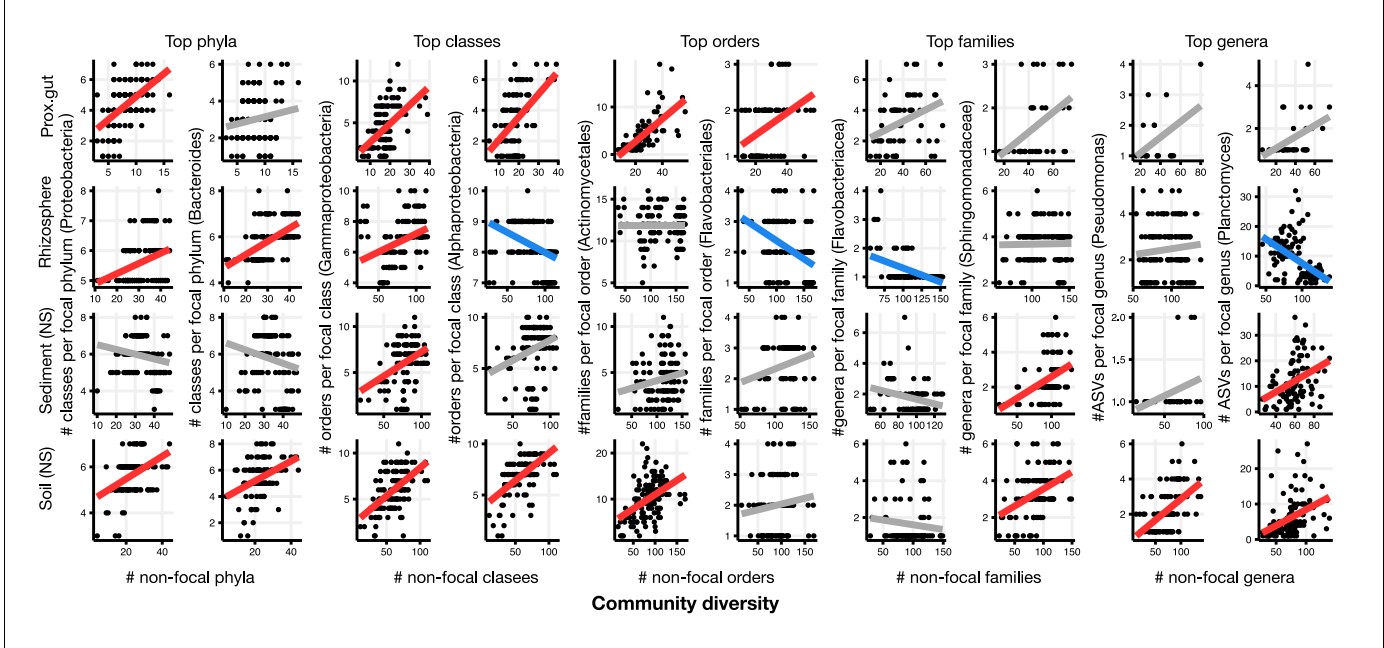

**Figure 2.** Focal-lineage diversity as a function of community diversity in the top two most prevalent taxa at each taxonomic level. As in *Figure 1*, the x-axes show community diversity in units of the number of non-focal taxa (e.g. the number of non-Proteobacteria phyla for the left-most column), and the y-axes show the taxonomic ratio within the focal taxon (e.g. the number of classes within Proteobacteria). Significant positive diversity slopes are shown in red, negative in blue (linear models, p<0.05, Bonferroni corrected for 17 tests), and non-significant in grey. Note that linear models are distinct from GLMMs, and are for illustrative purposes only. Four representative environments are shown (see *Figure 2—figure supplement 2–16* for plots in all 17 environments).

The online version of this article includes the following figure supplement(s) for figure 2:

**Figure supplement 1.** Distributions of diversity slope estimates across different random effects, from the GLMMs predicting focal lineage diversity as a function of community diversity.

**Figure supplement 2.** Focal-lineage diversity as a function of community diversity across biomes in Proteobacteria.

**Figure supplement 3.** Focal-lineage diversity as a function of community diversity across biomes in Bacteroidetes.

**Figure supplement 4.** Focal-lineage diversity as a function of community diversity across biomes in Actinobacteria.

**Figure supplement 5.** Focal-lineage diversity as a function of community diversity across biomes in Gammaproteobacteria.

**Figure supplement 6.** Focal-lineage diversity as a function of community diversity across biomes in Alphaproteobacteria.

**Figure supplement 7.** Focal-lineage diversity as a function of community diversity across biomes in Actinobacteria.

**Figure supplement 8.** Focal-lineage diversity as a function of community diversity across biomes in Actinomycetales.

**Figure supplement 9.** Focal-lineage diversity as a function of community diversity across biomes in Flavobacteriales.

**Figure supplement 10.** Focal-lineage diversity as a function of community diversity across biomes in Rhizobiales.

**Figure supplement 11.** Focal-lineage diversity as a function of community diversity across biomes in Flavobacteriaceae.

**Figure supplement 12.** Focal-lineage diversity as a function of community diversity across biomes in Sphingomonadaceae.

**Figure supplement 13.** Focal-lineage diversity as a function of community diversity across biomes in Verrucomicrobiaceae.

**Figure supplement 14.** Focal-lineage diversity as a function of community diversity across biomes in Pseudomonas.

**Figure supplement 15.** Focal-lineage diversity as a function of community diversity across biomes in Planctomyces.

**Figure supplement 16.** Focal-lineage diversity as a function of community diversity across biomes in Clostridium.

**Figure supplement 17.** Null models based on Neutral Theory.

**Figure supplement 18.** Lineage diversity (mean ASV:Genus ratio among all lineages) as a function of community diversity (number of genera) in the EMP data.

**Figure supplement 19.** Taxonomic ratios estimated from simulated rarefied sequence data.

**Figure supplement 20.** Linear, quadratic, and cubic models for the relationship between focal-lineage diversity and community diversity for varying levels of % nucleotide identity.

**Figure supplement 21.** Focal clusters at 75% nucleotide identity.

**Figure supplement 22.** Focal clusters at 80% nucleotide identity.

**Figure supplement 23.** Focal clusters at 85% nucleotide identity.

**Figure supplement 24.** Focal clusters at 90% nucleotide identity.

**Figure supplement 25.** Focal clusters at 95% nucleotide identity.

**Figure supplement 26.** Focal clusters at 97% nucleotide identity.

**Table 2.** GLMMs applied to data simulated under null models.

Null models 1 and 2 were generated under the ZSM distribution, with a single distribution for the whole dataset (Model 1) or one distribution per environment (Model 2). Model 3 is similar to Model 1, except with a single Poisson distribution for the whole dataset, and +DBD or +EC refer to adding these effects to all ASVs in each sample (see Materials and methods and *Figure 2—figure supplement 17*). Each row reports the effect of community diversity (Div) on focal lineage diversity, as well as its standard error, Wald z-statistic for its effect size and the corresponding *P*-value (Wald test) (left section), or standard deviation on the slope for the significant random effects (right section). SE = standard error, Env = environment type, Lin = lineage type, Sample = EMP Sample ID. n.s. = not significant (likelihood-ratio test), n.t. = not tested, because separate environments were not included in Models 1 or 3.

| | Slope (fixed effects) | | | | Stand dev on the slope (random effects) | | | |
|---|---|---|---|---|---|---|---|---|
| | Div | SE | z | *P* | Env | Lin | Lin*Env | Sample |
| Model 1 | −**0.005** | 0.000 | −9.807 | **<2 e −16** | n.t. | 0.639 | n.t. | n.s. |
| Model 2 | n.s. | | | | | | | |
| Model 3 | −0.012 | 0.002 | −6.552 | **5.69e-11** | n.t. | 0.021 | n.t. | n.s. |
| Model 3 + DBD | **0.016** | 0.001 | 11.48 | **<2e-16** | n.t. | 0.008 | n.t. | n.s. |
| Model 3 + EC | −0.011 | 0.002 | −6.14 | **8.26e-10** | n.t. | ns | n.t. | n.s. |

## DBD reaches a plateau at high diversity

It is expected from theory and experimental studies that a positive DBD relationship should eventually reach a plateau, giving way to EC as niches become saturated (*Brockhurst et al., 2007*; *Gómez and Buckling, 2013*). This expectation is borne out in our dataset, particularly in the nucleotide sequence-based analyses which support quadratic or cubic relationships over linear diversity slopes (*Figure 2—figure supplement 20*). For example, in the animal distal gut, a relatively low-diversity biome, we observed a strong linear DBD relationship at most phylogenetic depths; in contrast, the much more diverse soil biome clearly reaches a plateau (*Figure 2—figure supplements 21–26*).

To comprehensively test the hypothesis that more diverse microbiomes experience weaker DBD due to saturated niche space, we used a GLMM including the interaction between diversity and environment as a fixed effect. We considered this model only for taxonomic ratios with significant diversity slope variation by environment (*Table 1*): Family:Order, Order:Class, and Class:Phylum. Diversity slopes were significantly higher in less diverse (often host-associated) biomes, suggesting that niche filling leads to a plateau of DBD in more diverse biomes (*Figure 3*, *Supplementary file 1* Section 3). The interaction observed in the real EMP data between community diversity and biome type in shaping focal lineage diversity was not observed under a neutral null (Model 2, in which each environment has its own characteristic level of diversity) (*Supplementary file 3* Section 1.2). The DBD plateau observed in more diverse biomes is thus not readily explained by a neutral model, nor is rarefaction expected to bias the diversity slope estimates, particularly at the Class:Phylum level

**Table 3.** GLMMs with community diversity measured using Shannon diversity.

Results are shown from GLMMs with Shannon diversity of non-focal taxa (Div) as a predictor of ASVs richness of focal taxa. Each row reports the estimate (Div), as well as its standard error, Wald z-statistic for its effect size and the corresponding *P*-value (Wald test) (left section), or standard deviation on the slope for the significant random effects (right section). SE = standard error, Env = environment type, Lin = lineage type, Lab = Principal Investigator ID, Sample = EMP Sample ID. n.s. = not significant (likelihood-ratio test).

| Fixed effects | | | | | Random effects | | | | |
|---|---|---|---|---|---|---|---|---|---|
| | Div | SE | z | p | Env | Lin | Env*Lin | Env*Lab | Sample |
| Genus | 0.055 | 0.013 | 4.33 | 1.49e-05 | n.s. | 0.08 | 0.15 | 0.085 | 0.054 |
| Family | 0.148 | 0227 | 6.491 | 8.51e-11 | n.s. | 0.184 | 0.268 | 0.16 | 0.134 |
| Order | 0.378 | 0.038 | 9.864 | <2e-16 | n.s. | 0.34 | 0.417 | 0.258 | 0.202 |
| Class | 0.398 | 0.05 | 7.973 | 1.54e-15 | n.s. | 0.369 | 0.46 | 0.326 | 0.262 |
| Phylum | 0.319 | 0.088 | 3.614 | 0.0003 | 0.169 | 0.316 | 0.5 | 0.495 | 0.378 |

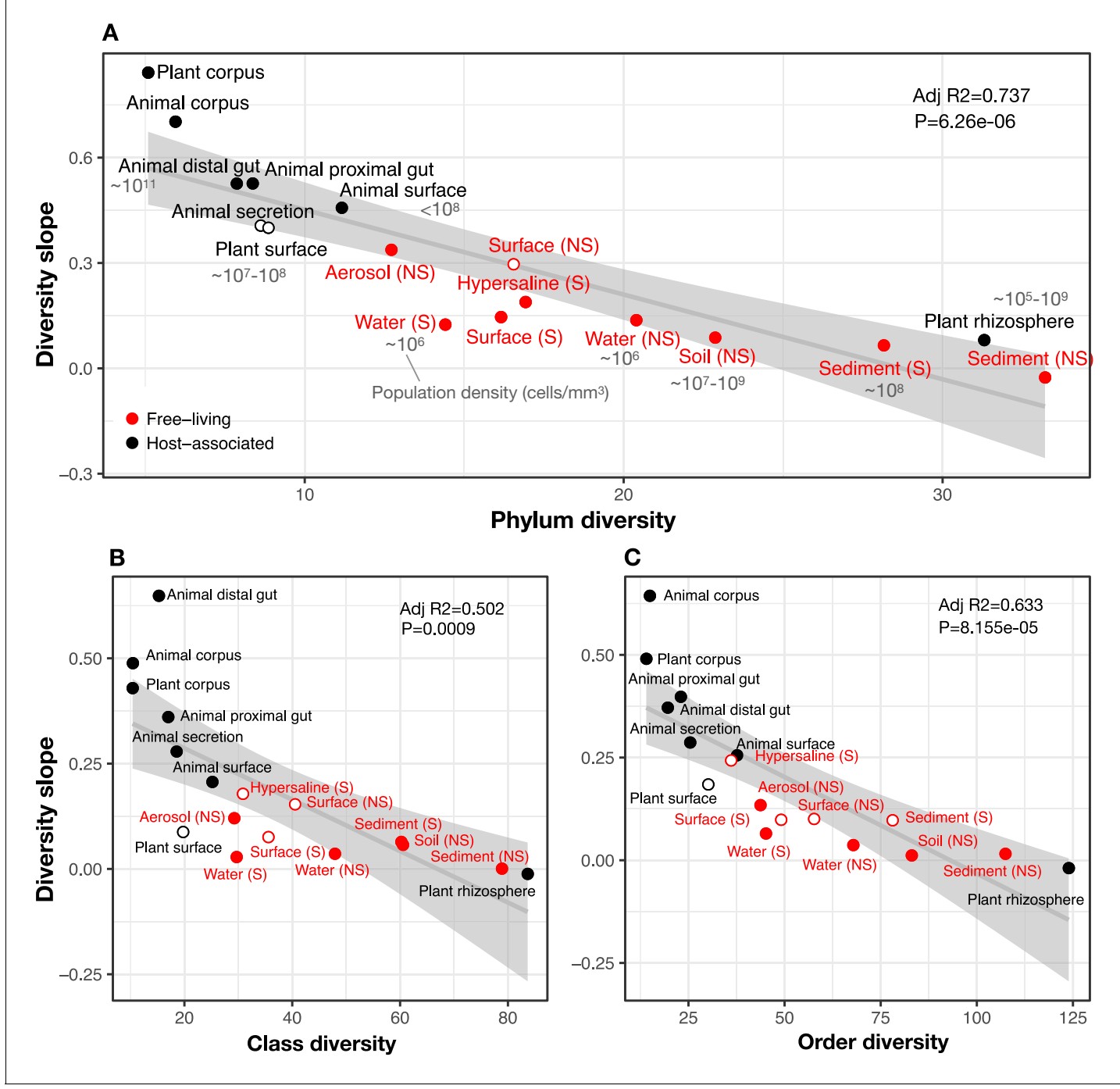

**Figure 3.** The diversity slope of focal taxa is higher in low-diversity (often host-associated) microbiomes. The x-axis shows the mean number of non-focal taxa: (A) phyla, (B) classes, and (C) orders in each biome. On the y-axis, the diversity slope was estimated by a GLMM predicting focal lineage diversity as a function of the interaction between community diversity and environment type at the level of (**A**) Class:Phylum, (**B**) Order:Class, and (**C**) Family:Order ratios (*Supplementary file 1* Section 3). The line represents a linear regression; the shaded area depicts 95% confidence limits of the fitted values. Adjusted $R^2$ and P-values from the linear fits are shown at the top right of each panel. See *Supplementary file 2* for model goodness of fit. Slopes not significantly different from zero are shown as empty circles. Estimates of bacterial cell density from the literature are indicated in grey text, in units of bacteria/$mm^3$. For animal (skin) and plant surface, units of bacteria/$mm^2$ were converted to $mm^3$ assuming layers of bacteria one micron thick. For rhizosphere samples we assume a density of 1–2 g/$cm^3$ (*Kennedy and de Luna, 2005*).

(*Figure 2—figure supplement 19*). This suggests that the plateau of DBD at higher levels of community diversity is not an artefact of data structure or sampling effort. Finally, we considered whether variation along the EC-DBD continuum could be explained by differential cell density across environments, which could affect both the frequency of cell-cell interactions (a biological effect) or the sampling depth (a technical artefact). Although precise estimates of cell densities in all EMP biomes are not available, we extracted plausible ranges for eight biomes from the literature (*Kennedy and de Luna, 2005*; *Lindow and Brandl, 2003*; *Sender et al., 2016*; *Whitman et al., 1998*) and annotated these in *Figure 3*. It is clear from this figure that relatively high- and low-density samples are found along the range of community taxonomic diversities, demonstrating that cell density is unlikely to drive the trend of decreasing diversity slopes with increasing community diversity.

## Abiotic drivers of diversity

Our results thus far suggest that community diversity is a major determinant of the EC-DBD continuum, and by extension that biotic interactions may override abiotic factors in determining where a community lies on the continuum. To formally test for the additional role abiotic drivers might play in generating the observed EC-DBD continuum, we analysed two data sets in more detail.

First, we analysed a subset of 192 EMP samples with measurements of four key abiotic factors shown to affect microbial diversity (pH, temperature, latitude, and elevation; *Delgado-Baquerizo et al., 2018*; *Lauber et al., 2009*; *Power et al., 2018*; *Schluter and Pennell, 2017*). We fitted a GLMM with focal lineage-specific diversity as the dependent variable, and with the number of non-focal lineages, the four abiotic factors and their interactions as predictors (fixed effects). As in the full EMP dataset (*Table 1*), focal lineage diversity was positively associated with community diversity at all taxonomic ratios in the EMP subset (*Table 4*). As expected, certain abiotic factors, alone or in combination with diversity, had significant effects on focal lineage diversity (*Table 4*).

**Table 4.** Community diversity has a stronger effect than abiotic factors on focal lineage diversity (EMP dataset).

Results are shown from GLMMs with community diversity (Div), four abiotic factors (temperature, elevation, pH, and latitude), and their interactions with community diversity, as predictors of focal lineage diversity. Random effects on the intercept included environment, lineage, lab ID and sample ID. Each row reports the taxonomic ratio, the predictors used in the GLMM (fixed effects only), their slope estimate (Est), standard error (SE) and *P*-value (P) (Wald test). Interactions are denoted as '*'. Random effects are not shown.

| | Predictor | Est | SE | P |
|---|---|---|---|---|
| ASV:Genus | Div | 0.128 | 0.013 | <2e-16 |
| | Temperature | 0.04 | 0.014 | 0.00479 |
| | Div*Temperature | 0.043 | 0.014 | 0.00175 |
| | Div*Latitude | 0.031 | 0.013 | 0.02119 |
| | Div*Elevation | −0.031 | 0.014 | 0.02829 |
| Genus:Family | Div | 0.094 | 0.009 | <2e-16 |
| | Temperature | 0.026 | 0.009 | 0.00268 |
| | pH | −0.042 | 0.009 | 5.88e-06 |
| Family:Order | Div | 0.131 | 0.01 | <2e-16 |
| Order:Class | Div | 0.184 | 0.01 | <2e-16 |
| | Div*Temperature | 0.032 | 0.009 | 0.000827 |
| | Div*Latitude | 0.023 | 0.008 | 0.005403 |
| Class:Phylum | Div | 0.236 | 0.011 | <2e-16 |
| | Div*Temperature | 0.059 | 0.014 | 2.15e-05 |
| | Div*Latitude | 0.03 | 0.011 | 0.00884 |

However, the effects of abiotic factors were always weaker than the effect of community diversity (*Table 4*; *Supplementary file 1* Section 4).

Second, we used a global 16S sequencing dataset of 237 soil samples associated with more detailed environmental metadata (*Delgado-Baquerizo et al., 2018*) which we reprocessed to yield ASVs comparable to those in the EMP (Materials and methods). This dataset revealed weaker evidence for DBD and stronger effects of abiotic variables on diversity. Community diversity generally had significant positive effects on focal-lineage diversity, but the effect was weak and not detectable at all taxonomic ratios (*Table 5*). Known abiotic drivers of soil bacterial diversity such as pH (*Lauber et al., 2009*) and latitude (*Delgado-Baquerizo et al., 2018*) had effects of similar or stronger magnitude compared to the effect of community diversity (*Table 5*, *Supplementary file 4*). The relatively weak effect of DBD and strong effect of abiotic drivers on diversity in this soil dataset can be explained by the fact that soils generally are highly diverse and have relatively low-diversity slopes (*Figure 3*).

We note that it remains possible that unmeasured abiotic effects could explain some of the DBD effects observed in the EMP. Although only a small subset of abiotic factors was considered, the

**Table 5.** GLMMs applied to a soil dataset.

Each row reports the taxonomic ratio, the predictors used in the GLMM (fixed effects only), their estimate (Est), standard error (SE) and *P*-value (P) (Wald test). Left columns: GLMM with community diversity (Div) and all abiotic variables considered separately, as predictors of focal lineage diversity. Right columns: GLMM with community diversity (Div) and the three first principle components (PCs) representing abiotic variables, as predictors of focal lineage diversity. n.s., non-significant (LRT test). All models provide a significantly better fit than null models without fixed effects (ΔAIC >10 and p<0.05; *Supplementary file 2*), except for the GLMM with abiotic factors at the Family:Order level, where latitude has a significant effect on focal lineage diversity but its effect is nearly null, with a ΔAIC between full and null model of 4 and a null marginal $R^2$.

| | GLMMs with abiotic variables | | | | GLMMs with the 3 first PCs | | | |
|---|---|---|---|---|---|---|---|---|
| | Predictor | Est | SE | P | Predictor | Est | SE | P |
| ASV:Genus | Div | n.s. | | | Div | 0.064 | 0.016 | 9.47e-05 |
| | Latitude | 0.294 | 0.025 | <2e-16 | PC1 | −0.065 | 0.007 | <2e-16 |
| | UV_light | −0.177 | 0.016 | <2e-16 | PC2 | −0.03 | 0.006 | 1.98e-05 |
| | MDR | 0.028 | 0.006 | 7.12e-06 | | | | |
| | NPP2003_2015 | −0.066 | 0.005 | <2e-16 | | | | |
| | Latitude2 | −0.3 | 0.029 | <2e-16 | | | | |
| | Clay_silt2 | −0.012 | 0.004 | 0.003 | | | | |
| | Soil_N2 | −0.007 | 0.001 | 1.66e-06 | | | | |
| | Soil_C_N_ratio2 | 0.003 | 0.001 | 0.004 | | | | |
| | PSEA2 | 0.01 | 0.002 | 4.84e-06 | | | | |
| | MDR2 | 0.017 | 0.003 | 2.40e-08 | | | | |
| | NPP2003_20152 | −0.016 | 0.004 | 0.0001 | | | | |
| Genus:Family | Div | 0.032 | 0.01 | 0.0011 | Div | 0.033 | 0.01 | 0.001 |
| | Latitude | −0.035 | 0.006 | 2.04e-09 | PC1 | −0.016 | 0.006 | 0.02 |
| | | | | | PC2 | 0.02 | 0.006 | 0.00089 |
| Family:Order | Div | n.s. | | | Div | n.s. | | |
| | Latitude | −0.0005 | 0.0002 | 0.0105 | PC1 | −0.026 | 0.007 | 0.00032 |
| | | | | | Div*PC1 | 0.04 | 0.006 | 2.14e-12 |
| | | | | | Div*PC3 | 0.023 | 0.005 | 1.68e-06 |
| Order:Class | Null model with no predictor was significant | | | | | | | |
| Class:Phylum | Div | 0.032 | 0.01 | 0.00174 | Div | 0.032 | 0.01 | 0.003 |
| | pH | 0.074 | 0.01 | 4.37e-13 | PC1 | −0.051 | 0.01 | 3.54e-07 |
| | | | | | PC2 | −0.028 | 0.01 | 0.006 |

generally positive diversity slopes in the EMP are not likely to be driven by these factors in the abiotic environment (*Table 4*). Specifically, we consider it unlikely that unmeasured abiotic factors would always act similarly, and in the same direction across multiple different environments, to drive DBD. However, as demonstrated in soil (*Table 5*), abiotic factors may become increasingly important in highly diverse biomes with weak DBD.

## DBD is more pronounced in resident taxa than in migrants or generalists

A recent meta-analysis of 16S sequence data from a variety of biomes suggests there is an important distinction between generalist lineages found in many environments, compared to specialists with a more restricted distribution (*Sriswasdi et al., 2017*). Generalists were inferred to have higher speciation rates, suggesting that the DBD-EC balance might differ between generalists and specialists (*Sriswasdi et al., 2017*). To further investigate this difference, we defined 'resident', taxa with a strong preference for a specific biome, in addition to generalists without a strong biome preference in the EMP dataset. We first clustered environmental samples by their genus-level community composition using fuzzy *k*-means clustering (*Figure 4a*), which identified three major clusters: 'animal-associated', 'saline', and 'non-saline'. The clustering included some outliers (*e.g.* plant corpus grouping with animals), but was generally consistent with known distinctions between host-associated vs. free-living (*Thompson et al., 2017*), and saline vs. non-saline communities (*Auguet et al., 2010*; *Lozupone and Knight, 2007*). Resident genera were defined as those with a strong preference for a particular environment cluster (whether due to dispersal limitation or narrow niche breadth) using indicator species analysis (permutation test, p<0.05; *Figure 4a*; *Figure 4—figure supplement 1*; *Supplementary file 5*), and genera without a strong preference were considered generalists. When residents of one environmental cluster were (relatively infrequently) observed in a different cluster, we defined them as 'migrants' in that sample. For each environment cluster, we ran a GLMM with resident genus-level diversity (the number of non-focal genera) as a predictor of focal-lineage diversity (the ASV:Genus ratio) for residents, generalists, or migrants to that sample (*Supplementary file 1* Section 5).

Resident community diversity had no significant effect on the diversity of generalists in animal-associated, saline and non-saline clusters (GLMM, Wald test, p>0.05), but was positively correlated with lineage-specific resident diversity (GLMM, Wald test, z = 7.1, p=1.25e-12; z = 3.316, p=0.0009; z = 7.109, p=1.17e-12, respectively). Resident community diversity significantly decreased migrant diversity in saline (GLMM, z = −3.194, p=0.0014) and non-saline environment clusters (GLMM, z = −2.840, p=0.0045), but had no significant effect in the animal-associated cluster (GLMM, p>0.05) (*Figure 4b*). These results suggest that, although generalist lineages may have higher speciation rates and colonize more habitats than specialists (*Sriswasdi et al., 2017*), they have lower diversity slopes. Migrants to the 'wrong' environment experience even less DBD, and are even subject to EC in two out of three environment types (*Figure 4b*). The accumulation of diversity via successful establishment of migrants may thus be limited, presumably because most niches are already occupied by residents.

## Discussion

Using ~10 million individual marker sequences from the EMP, we demonstrate an overall trend for diversity in focal lineages to be positively associated with overall community diversity, albeit with significant variation across lineages and environments. The strength of the DBD relationship dissipates with increasing microbiome diversity, which we hypothesize is caused by niche saturation. In more diverse biomes such as soil, abiotic factors therefore may become relatively more important in driving focal-lineage diversity. The effect of DBD is strongest among habitat specialists (residents), suggesting that long-term niche adaptation tends to select against the establishment of migrant diversity.

While most of the DBD literature considers a model of evolutionary diversification (*Schluter and Pennell, 2017*; *Whittaker, 1972*), our results pertain mainly to ecological community assembly dynamics. At the limited resolution of 16S rRNA gene sequences, we do not expect measurable diversification within an individual microbiome sample (*Kuo and Ochman, 2009b*); however, community diversity could still select for (as in DBD) or against (as in EC) increasing diversity in a focal

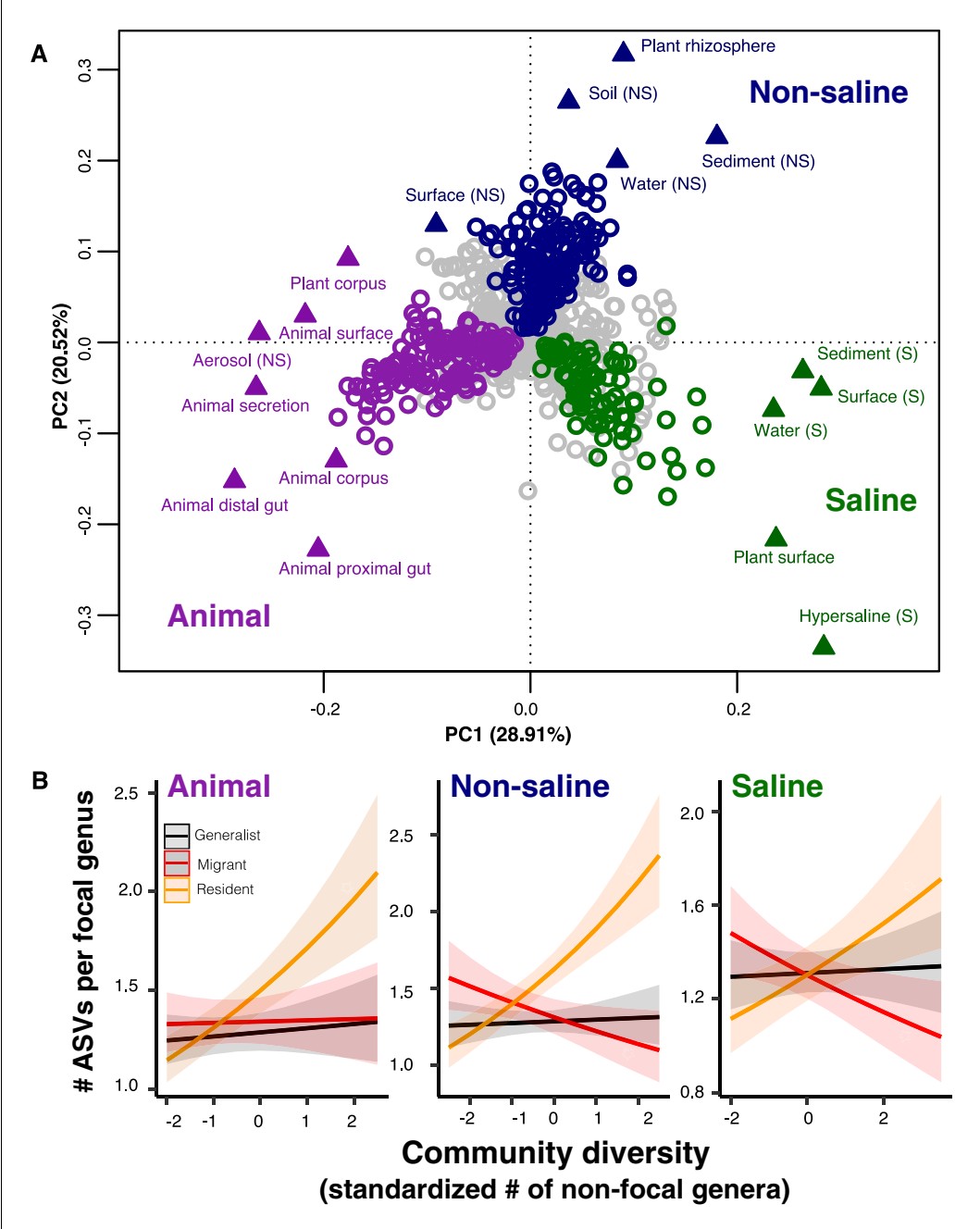

**Figure 4.** The DBD relationship varies between resident and non-resident genera. (**A**) Ordination showing genera clustering into their preferred environment clusters. The matrix of 17 environments (rows) by 1128 genera (columns) by, with the matrix entries indicating the percentage of samples from a given environment in which each genus is present, was subjected to principal components analysis (PCA). Circles indicate genera and triangles indicate environments (EMPO 3 biomes). coloured circles are genera inferred by indicator species analysis to be residents of a certain environmental cluster, and grey circles are generalist genera. The three environment clusters identified by fuzzy *k*-means clustering are: Non-saline (NS, blue), saline (S, green) and animal-associated (purple). Triangles of the same colour indicate EMPO 3 biomes clustered into the same environmental cluster. (**B**) DBD in resident versus non-resident genera across environment clusters. Results of GLMMs modelling focal lineage diversity as a function of the interaction between community diversity and resident/migrant/generalist status. The x-axis shows the standardized number of non-focal resident genera (community diversity); the y-axis shows the number of ASVs per focal genus. Resident focal genera are shown in orange, migrant focal genera in red, and generalist focal genera in black. Red stars indicate a significantly positive or negative slope (Wald test, p<0.005). See *Supplementary file 2* for model goodness of fit.

The online version of this article includes the following figure supplement(s) for figure 4:

**Figure supplement 1.** Resident genera of environment clusters.

lineage, even if this lineage diversified before the sampled community assembled. Future work with higher resolution genomic or metagenomic data will enable testing if and how DBD arises in microbial communities via evolutionary diversification, and also how prokaryote diversification is affected by other community members including phages (*Brockhurst et al., 2005*), protists (*Meyer and Kassen, 2007*), and fungi (*Kastman et al., 2016*). Predator-prey, cross-feeding, and other biotic interactions with these non-prokaryotic community members could explain some of the unaccounted variation we observed in diversity slopes across environments.

Our dataset also provides an opportunity to explore how DBD relates to genome size evolution. Bacteria with larger repertoires of accessory genes, and thus larger genomes, are able to occupy a wider range of niches (*Barberán et al., 2014*). Taxa with larger genomes might therefore be hypothesized to better survive and thrive when they disperse into a new location, exhibiting stronger DBD. Although a comprehensive test of this hypothesis will require higher resolution genomic or metagenomic data, as a preliminary exploration we assigned genome sizes to 576 focal genera for which at least one whole genome sequence was available (using the largest recorded genome size for each genus) and added an interaction term between genome size and diversity as a fixed effect in the GLMM (Materials and methods). Consistent with our expectation, we observed a significant positive effect of genome size on the diversity slope (GLMM, Wald test, $z = 2.5$, $p=0.01$; *Figure 5*, *Supplementary file 1* Section 6). This effect was not observed in null models, in which the interaction between community diversity and focal genus genome size was never significant (*Supplementary file 3* Section 1.3 and 2.2) and so this effect of genome size cannot be trivially explained by data structure. The positive relationship between genome size and DBD is likely even stronger than estimated, because assigning genome sizes to entire genera is imprecise (i.e. there is variation in genome size within a genus, or even within species), therefore weakening the correlation.

The positive correlation between genome size and DBD observed here could be driven by larger metabolic repertoires encoded by larger genomes (*Barberán et al., 2014*), potentially creating more opportunities to benefit from cross-feeding, niche construction (*San Roman and Wagner, 2018*), and other interspecies interactions. This tendency appears to be at odds with the Black Queen hypothesis, which predicts that social conflict between interacting species leads to the inactivation and loss of genes involved in shareable metabolites (public goods), eventually resulting in reduced genome size (*Morris et al., 2012*). Such a process would produce a negative correlation between the degree of species interactions (i.e. community diversity) and genome size (*Morris et al., 2012*). The interaction between genome size, biotic interactions and diversification thus deserves further study.

Alongside theory and experimental data, the EMP survey data provide a window into the biotic drivers of microbial diversity in nature. In particular, our correlational results support previous experiments and theory showing that DBD is strong when community diversity is low (*Calcagno et al., 2017*; *Jousset et al., 2016*), driving the accumulation of diversity in a positive feedback loop until niches are filled and EC starts

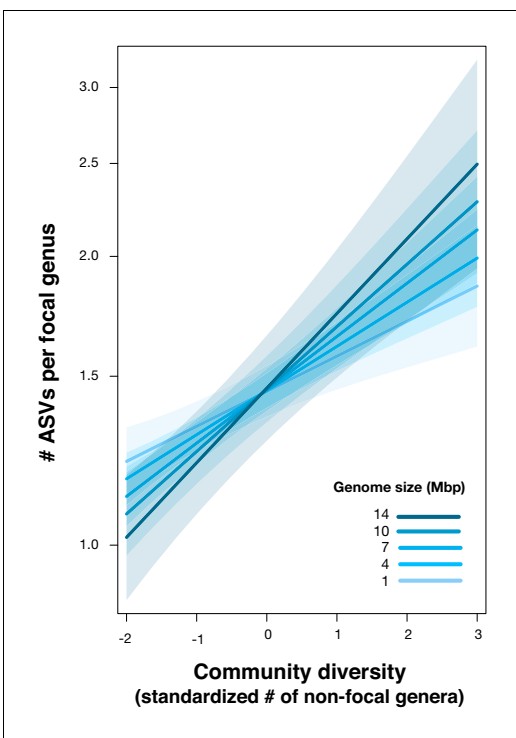

**Figure 5.** Positive effect of genome size on DBD. Results are shown from a GLMM predicting focal lineage diversity as a function of the interaction between community diversity and genome size at the ASV:Genus ratio (*Supplementary file 1* Section 6). The x-axis shows the standardized number of non-focal genera (community diversity); the y-axis shows the number of ASVs per focal genus. Variable diversity slopes corresponding to different genome sizes are shown in a blue colour gradient; the shaded area depicts 95% confidence limits of the fitted values. See *Supplementary file 2* for model goodness of fit.

to predominate (*Bailey et al., 2013*; *Brockhurst et al., 2007*; *Gómez and Buckling, 2013*; *Meyer and Kassen, 2007*). However, due to the correlational nature of the EMP data, it is not possible to test whether DBD is primarily due to the creation of novel niches via biotic interactions and niche construction (*Laland et al., 1999*), or due to increased competition leading to specialization on underexploited resources (*Hibbing et al., 2010*; *Jousset et al., 2016*). We hope future higher resolution genomic studies, and complementary experiments, will be able to elucidate the types of biotic interactions that promote microbiome diversity. Regardless of the underlying mechanisms, our results demonstrate a general scaling between different levels of community diversity, which has important implications for modelling and predicting community function and stability in response to perturbations (*Coyte et al., 2015*; *Pennekamp et al., 2018*). The answer to the question 'why are microbiomes so diverse?' might in a large part be because microbiomes are so diverse (*Emerson and Kolm, 2005*).

## Data and materials availability

All data is available from the Earth Microbiome Project (ftp://ftp.microbio.me/), as detailed in the Methods. All computer code used for analysis are available at (https://github.com/Naima16/dbd.git; *Madi, 2020*; copy archived at swh:1:rev:ecb4f844264b72eaa8cbd708244ecd32d414c7dd).

# Materials and methods

## Earth Microbiome Project dataset

We used the EMP '2000 subset' of 16S rRNA gene sequences, rarefied to 5000 sequences per sample. This subset contains 155,002 ASVs from 2000 samples with an even distribution across 17 natural environments (EMP Ontology level 3). Data were downloaded from the EMP FTP server (ftp://ftp.microbio.me/), on February 9, 2018.

Specifically, 16S rRNA-V4 region reads (90 bp, GreenGenes 13.8 taxonomy) along with environmental data and EMPO3 designations (http://press.igsb.anl.gov/earthmicrobiome/protocols-and-standards/empo/) were used. Sequence summaries were downloaded from: ftp://ftp.microbio.me/emp/release1/otu_distributions/otu_summary.emp_deblur_90bp.subset_2k.rare_5000.tsv, environmental data from: ftp://ftp.microbio.me/emp/release1/mapping_files/emp_qiime_mapping_release1.tsv, and EMPO3 designations from: ftp://ftp.microbio.me/emp/release1/mapping_files/emp_qiime_mapping_subset_2k.tsv.

The list of the associated 97 studies and 61 corresponding principal investigator identities were downloaded from https://www.nature.com/articles/nature24621#s1.

Based on the ASV annotations across samples, we estimated the taxonomic ratio for each focal lineage (ASV:Genus, Genus:Family, Family:Order, Order:Class and Class:Phylum), along with the number of non-focal lineages (dbd_analys_input.py, glmm_analys_input.py, Python Version 2.7). Unclassified ASVs were removed from the analyses.

## Generalized linear mixed model (GLMM) analyses

We used GLMMs to determine how focal lineage diversity (e.g. its ASV:Genus ratio) is affected by community diversity (e.g. non-focal genera). The effects of environment (as defined by the EMP Ontology 'level 3 biomes') and the focal lineage identity were included as random effects on the slope and intercept. We also controlled for the submitting laboratory (identified by the principal investigator) and the EMP unique sample identifier (i.e. if two taxa were part of the same sample).

All models were fitted in Rstudio (Version 1.1.442, R Version 3.5.2) using the glmer function of the lme4 package (*Bates et al., 2015*). Data standardization (transformation to a mean of zero and a standard deviation of one) was applied to all predictors to get comparable estimates. In models with only one predictor, applying standardization resolved convergence warnings and considerably sped up the optimization. We first tested the significance of random effects, by using likelihood-ratio tests (LRTs, implemented in the anova function in the R stats package) on nested models where each random effect was dropped one at a time. We then assessed the significance of fixed effects using the drop1 function from the stats package with the likelihood-ratio test option (this function drops individual terms from the full model and compares models based on the AIC). We calculated the Akaike information criterion (AIC) of each significant model and a null model including all random effects

but no fixed effects other than the intercept. We then report the difference in AIC between the full and null models (ΔAIC), along with a likelihood-ratio test *p*-value to assess the significance of the full model relative to the null. Only significant models (p<0.05) are reported.

As an additional test of the goodness of fit for the significant models, we estimated the coefficient of determination ($R^2$) using the r.squaredGLMM function from the MuMIn R package. This function implements a method developed by Nakagawa and Schielzeth and its extension for random slopes (*Johnson, 2014*; *Nakagawa and Schielzeth, 2013*). Two values were estimated: the marginal $R^2$, as a measure of the variance explained only by fixed effects, and the conditional $R^2$ as a measure of the variance explained by the entire model (both fixed effects and random effects). Only results from $R^2$ estimation based on lognormal and trigamma methods were reported because they are specific to the logarithmic link function used in all GLMMs.

Diagnostic plots (plot and qqnorm R functions in base and stats packages) were checked for each model to ensure that residual homoscedasticity (homogeneity of variance) was fulfilled: no increase of the variance with fitted values and residuals were symmetrically distributed tending to cluster around the 0 of the ordinate, but with an expected pattern due to count data. Normality plots were imperfect, but they generally showed that the residuals were close to being normally distributed. The assumption of normality is often difficult to fulfill with high numbers of observations, as is the case in our models (https://www.statisticshowto.datasciencecentral.com/shapiro-wilk-test/), and non-normality is less of concern than heteroscedastic for the validity of GLMMs (https://bbolker.github.io/mixedmodels-misc/ecostats_chap.html#diagnostics).

We tested for overdispersion using the overdisp_fun R function available at https://bbolker.github.io/mixedmodels-misc/glmmFAQ.html, and found that all the models were not overdispersed, but rather were underdispersed: the ratio of the sum of squared Pearson residuals to residual degrees of freedom was <1 and non-significant when tested with a chi-squared test. The only exception was Shannon diversity-based GLMMs. In case of underdispersion and given that underdispersion leads to more conservative results, we retained the GLMMs with Poisson error distribution, despite the underdispersion. (GLMM FAQ; Ben Bolker and others; 25 September 2018; https://bbolker.github.io/mixedmodels-misc/glmmFAQ.html#underdispersion). For Shannon diversity-based GLMMs, we accounted for overdispersion by adding an observation-level random effect to the GLMMs (*Elston et al., 2001*).

## Taxonomy-based GLMMs

To test how focal lineage diversity (e.g. its ASV:Genus ratio) is affected by community diversity (e.g. non-focal genera richness), for different environment types and lineages across all taxonomic ratios, we used generalized linear mixed models (GLMMs) fitted on the EMP dataset. As the dependent variable (focal lineage diversity, defined as taxonomic ratios, ASV:Genus, Genus:Family, Family:Order, Order:Class, and Class:Phylum) was a count response, we used a Poisson error distribution with a log link function. Community diversity (number of non-focal lineages: non-focal Genera, Families, Orders, Classes, and Phyla), standardized to a mean of zero and a standard deviation of one, was specified as the predictor (fixed effect). We included the following random effects on the slope and intercept: lineage (Lin), environment (Env), environment nested within lineage (a lineage may be present in different environments) and lab (the principal investigator who conducted the EMP study) nested within environment (different labs sampled and sequenced a given environment) (as suggested in http://bbolker.github.io/mixedmodels-misc/glmmFAQ.html). Defining random effects on the slope enabled us to test slope variation across groups of each categorical variable (*e.g.* slope variation between different environments or different lineages). We included the EMP unique sample ID as a random effect to control for dependencies between observations (if two taxa were part of the same sample) (*Table 1*, *Supplementary file 1* section 1).

## Shannon diversity-based GLMMs

We also tested whether ASV diversity in a focal taxon is dependent on the diversity of all other ASVs in that sample (rather than the diversity at only the focal taxonomic level, as in the taxonomy-based GLMMs above). We used the Shannon diversity index, which is robust to differences in sampling effort, and generally reaches a plateau at 5000 sequences or fewer. To do so, we fitted a GLMM with the number of ASVs per focal taxon as the response variable, and the Shannon diversity based

on ASVs across all non-focal taxa (z-standardized) as the predictor (fixed effect) The random effects were kept as in the taxonomy-based GLMMs, but we added an observation-level random effect to account for overdispersion (*Table 3*, *Supplementary file 1* section 2). To avoid dependence between the response and predictor variables, we used the rarefied ASV dataset (5,000 ASVs/sample as above) as the response variable, and the Shannon diversity calculated on unrarefied data from the same samples as the predictor.

## Null models

We considered three null models, all of which randomize the associations between ASVs within a sample, thus breaking any true biotic interactions. These null models were randomly generated matrices of the same size as the real EMP dataset, but based on a distribution that arises from the Neutral Theory of Biodiversity. Neutral Theory postulates that the biodiversity of a metacommunity is governed by independent random population dynamics across species. The aggregate behaviour is quantified by the fundamental biodiversity number $\theta$, such that $\theta = 2 J_M \upsilon$, where $J_M$ is the size of the metacommunity and $\nu$ is the speciation rate. Parametrized by $\theta$, the metacommunity zero-sum multinomial distribution (mZSM) was developed to obtain random samples of size $J$ (*Alonso and McKane, 2004*). We used this mZSM distribution (implemented with the *sads* package in R; http://search.r-project.org/library/sads/html/dmzsm.html) to generate the counts of the ASVs for each dataset in models 1 and 2. Model 1 assumes that the whole dataset follows the same species abundance distribution (SAD), characterized by a mZSM with $\theta = 50$. Model 2 assumes that each environment has its own SAD and thus all the samples of a single environment are assigned the same $\theta$ but are distinct across environments ($\theta$ was chosen uniformly between 1 and 100). The number of samples per environment were the same as the EMP dataset. To obtain similar mean counts as the real dataset, we set $J = 1000$ for both models 1 and 2, in order to vary $\theta$ from 1 to 100. These values are reasonable based on previous studies that estimated these parameters from microbiome data (*Li and Ma, 2016*). We included a down-sampling step to replicate the zero-inflated nature of the real dataset (on average there were only 96 ASVs per sample while there was a total of 22,014 ASVs in the entire EMP dataset). To replicate the sampling effect due to rarefaction, we first created a vector of all individuals from a single sample. We then selected 5000 individuals at random whose identities determined which ASVs were found in that sample. These neutrally-derived random matrices, null models 1 and 2, were plotted using the same plots (ASV:Genus vs number of genera) as the real EMP dataset and were then analysed using GLMMs with community diversity as a predictor of focal lineage diversity (fixed effect), with lineage identity and EMP sample ID as random effects. For Model 1, the slope was significantly negative (GLMM, Wald test, z=-9.807, *P*<2e-16). For Model 2, the null GLMM (including the intercept only) was significant, meaning that the community diversity has no significant effect on focal lineages diversity (Likelihood-ratio test between the model with the predictor and the intercept-only model, *P*=0.9399).

To generate a null model for a metacommunity assembled by niche processes, null model 3 was made by sampling from a single Poisson distribution ($\lambda = 0.01$) for each element of the data matrix. We used the Poisson distribution as a sensitivity analysis compared to the ZSM, and found the two behave quite similarly (i.e. Model 1 and 3 produce qualitatively similar results). The probability of size zero was sufficiently large that the down-sampling step was not needed for this model. Next, DBD and EC effects were added to null model 3 according to the following procedure. An element was chosen at random in a sample and tested if it is empty or full (i.e. check the presence/absence of a particular ASV). If the element is full then the DBD algorithm fills an empty element chosen at random in the same sample, while the EC algorithm empties a filled element in the same sample. This is to mimic the effect of DBD creating a niche for a new ASV, or EC removing a niche based on the existing diversity. The strength of DBD or EC effects were determined by the percent of elements tested. These data were analysed with GLMMs to test the power of our models to detect DBD or EC (*Table 2*, *Supplementary file 3* Section 2.1).

## Rarefaction simulation

We constructed a simple simulation in which each microbiome sample may differ in total diversity (*e.g.* in the observed range of genera) while maintaining a constant taxonomic ratio (e.g. ASV:genus ratio = 2). To mimic rarefaction, we then sampled a set number of sequencing reads from each

synthetic community, assuming ASVs are sampled with equal probability and plotted the observed taxonomic ratio (*Figure 2—figure supplement 19*). This simple simulation is implemented in permute_ASVs_synthetic.pl.

## Nucleotide sequence-based analysis

We clustered ASVs at decreasing levels of nucleotide identity, from 100% identical ASVs down to 75% identity (roughly equivalent to phyla [*Konstantinidis and Tiedje, 2005*]). We estimated focal cluster diversity as the mean number of descendants per cluster (e.g. number of 100% clusters per 97% cluster) and plotted this against the total number of clusters (97% identity in this example). This approach has the advantage of including sequences even if they come from unnamed taxa. For each of the six nucleotide divergence ratios tested, the relationship between total number of clusters and focal cluster diversity was positive (*Figure 2—figure supplement 20*), consistent with DBD and suggesting that the taxonomic analyses were qualitatively unbiased.

Fasta files with all ASVs per sample were produced by a python script (Construct_fasta_per_sample.py, Python Version 2.7) from the sequences summary file (otu_summary.emp_deblur_90 bp.subset_2 k.rare_5000 from EMP ftp server). We clustered sequences from each sample using USEARCH V9.2 and estimated sample diversity as the total number of clusters at a given level (e.g. 97% identity) and focal cluster diversity as the mean number of descendent clusters (e.g. number of 100% clusters per 97% cluster). To describe the putative DBD or EC relationships, we tested three models: linear, quadratic and cubic (lm function in R). Model comparisons were based on the adjusted $R^2$ (*Figure 2—figure supplement 20*).

We note that diversity at level $i$ ($d_i$) and at level $i+1$ ($d_{i+1}/d_i$) are not independent in this analysis because $d_{i+1}$ must be greater than or equal to $d_i$. To assess the effects of this non-independence on the results, we conducted permutation tests by randomizing the associations between $d_i$ and $d_{i+1}$. Using 999 permutations, *P*-values were calculated based on how many times we observed a correlation greater than that seen in the real data (cor.test R function with kendall method). In each permutation, we recalculated the significance test (Wald z) for the correlation in the randomized data, and then computed the *P*-value based on how many times we observed a z value greater than that of the original data. At all six levels of nucleotide identity, the real data always showed a significantly stronger positive correlation when compared to permuted data (p=0.001), indicating that the DBD patterns was not an artefact of the dependence structure in the data.

The effect of community diversity on focal cluster diversity was also tested across different environments analysed separately. We modelled this relationship with linear, quadratic and cubic fits, and compared those models based on the adjusted $R^2$ (*Figure 2—figure supplements 21–26*).

## DBD variation across environments

We tested the variation of focal lineage diversity slopes across different environments by including EMPO 3 biome type as a fixed effect. We fitted a GLMM with the interaction between community diversity and environment type as a predictor of focal lineage diversity. All other random effects on intercept and slope were kept as in the previous GLMMs (*Figure 3*, *Supplementary file 1* Section 3). DBD variation across environments was tested for Family:Order, Order:Class and Class:Phylum taxonomic ratios, as diversity slope variation by environment was statistically significant (likelihood-ratio test, p<0.05) for these ratios in the taxonomy-based models (*Table 1*).

## Abiotic effects

To test for the relative effect of biotic and abiotic environmental variables on focal lineage diversity across different taxonomic ratios, we used a separate GLMM, with Poisson error distribution and a log link function, for every ratio. We fitted the GLMM on a subset (~10%) of the whole dataset, 192 samples (from water: saline (19) and non-saline (44), surface: saline (42) and non-saline (19), sediment: saline (22) and non-saline (31), soil (8) and plant rhizosphere (7)), for which measurements of four key abiotic variables (temperature, pH, latitude and elevation) were available. As predictors of focal lineage diversity (fixed effects), we included non-focal community diversity and abiotic variables, as well as their interactions. All predictors were standardized to a mean of zero and a standard deviation of one to obtain comparable estimates. The GLMM had the same random effects as in the previous analysis, but only on the intercept for simplicity (*Table 4*, *Supplementary file 1* section 4).

## Soil dataset analysis

We used the *Delgado-Baquerizo et al., 2018* soil microbiome survey (237 samples from 18 countries) to further test the relative impacts of biotic versus abiotic drivers of diversity. Raw data and abiotic measurements were downloaded from Figshare (https://figshare.com/s/82a2d3f5d38ace925492; DOI: 10.6084/m9.figshare.5611321). 16S bioinformatic processing was performed using QIIME2 and Deblur with the same protocol as in *Thompson et al., 2017*. Raw data 16S rRNA gene (V3-V4 region), were processed by trimming the primers (341F/805R primer set) with qiime cutadapt trim-paired, then merged using qiime vsearch join-pairs. Sequences were quality filtered and denoised using Deblur with a trimming length of 400 bp. The resulting 400 bp Deblur BIOM table was filtered to keep only ASVs with at least 25 reads total over all samples and rarefied to a depth of 5000. Taxonomy was assigned with a Naive Bayes classifier trained on the V4-V3 region of 99% OTU Greengenes 13.8 sequences with qiime feature-classifier. We obtained a final dataset of 186 samples and 24,252 ASVs which was used as input for all statistical analysis as in the EMP dataset analysis. This data set included 14 environmental factors: aridity index (Aridity_Index), minimum and maximum temperature (MINT and MAXT), precipitation seasonality (PSEA), mean diurnal temperature range (MDR), ultra-violet (UV) radiation (UV_Light), net primary productivity (NPP2003_2015), soil texture (Clay_silt), pH; total C (Soil_C), N (Soil_N) and P (Soil_P) concentrations, C:N ratio (Soil_C_N_ratio) and Latitude.

We used a separate GLMM with Poisson error distribution and a log link function to test for the effect of biotic (non-focal community diversity) and abiotic environmental variables on focal lineage diversity (e.g. the ASV:Genus ratio for a focal genus), across different taxonomic ratios. We defined non-focal taxa diversity and abiotic variables as predictors (fixed effects) and the lineage identity as a random effect.

We also fitted the same model but with the first three principal components (PCs) from the principal component analysis (PCA, rda function, vegan R package) of the abiotic variables (a matrix of 237 samples (rows) by 14 abiotic variables (columns)), as well as the interactions between diversity and each PC, and the interaction between PCs as predictors (fixed effects).

Because of possible non-linear relationships between abiotic variables and diversity, GLMMs were fitted with a linear and a quadratic term for every abiotic variable. The quadratic terms were not significant, except for the ASV:genus ratio (*Table 5*; likelihood-ratio test, $p<2.2e-16$). The interaction terms were not significant except the interaction between diversity and PCs at Family:Order ratio (likelihood-ratio test, $p=2.182e-05$; *Table 5*, *Supplementary file 4*).

## Defining residents, generalists, and migrants

We defined a genus-level community composition matrix as a matrix of 17 environments (rows) by 1128 genera (columns), with the matrix entries indicating the percentage of samples from a given environment in which each genus is present. We clustered the environmental samples based on their genus-level community composition using fuzzy *k*-means clustering. The clustering (cmeans function, package e1071 in R) was done on the 'hellinger' transformed data (decostand function, vegan R package). To identify resident genera to each cluster, we used indicator species analysis (*Dufrene and Legendre, 1997*) as implemented in the indval function (labdsv R package). We defined residents as genera with indval indices between 0.4 and 0.9, with permutation test $p<0.05$. Genera not associated with any cluster were considered generalists. We used principal component analysis (PCA) on the community composition matrix to visualize the clustering and the indicator genera (rda function, vegan R package) (*Figure 4*). We then ran a separate GLMM for each environmental cluster, with resident genus-level diversity (number of non-focal genera) as a predictor of focal genus diversity (ASV:Genus ratio) for resident, migrant (residents of one cluster found in a different cluster) and generalist genera. The fixed effect was specified as the interaction between diversity and a factor defining the genus-cluster association (with three levels: resident, migrant and generalist). Random effects on intercept and slope were kept as in the GLMMs described above.

## Genome size analysis

We chose a subset of genera represented by one or more sequenced genomes in the NCBI microbial genomes database (https://www.ncbi.nlm.nih.gov/genome/browse#!/prokaryotes/). For these genera, a representative genome size was assigned by selecting the genome with the lowest

number of scaffolds (if no closed genomes were available) (*Supplementary file 6*). If multiple genomes were available with the same level of completion, the largest genome size was used, as smaller genomes could be artefacts of incomplete assembly which would bias the mean and median downward. Moreover, given the deletional bias in bacterial genomes (*Kuo and Ochman, 2009a*), the largest genome is likely more reflective of the ancestral genome size of the genus. Only genera with two or more ASVs in at least one sample were included in the analysis. Intracellular symbionts were excluded. We fitted a GLMM on the subset of data with known genome size (576 genera, ranging from ~1 to 15 Mbp) with the interaction between community diversity and genome size as a predictor of focal lineage diversity at the ASV:Genus level. All the other random effects on intercept and slope were kept as in the previous GLMMs (*Supplementary file 1* section 6).

## Acknowledgements

We thank Luke Thompson for assistance obtaining EMP data and Zofia Ecaterina Taranu, Vincent Fugère and Guillaume Larocque for advice on GLMMs. We are also grateful to Steven Kembel, Tom Battin, the reviewers Eric Kemen and Benjamin E Wolfe, and the editor Detlef Weigel for critical comments that improved the manuscript. Funding: This project was made possible by an NSERC Discovery Grant and Canada Research Chair to BJS.

## Additional information

### Funding

| Funder | Author |
| --- | --- |
| Natural Sciences and Engineering Research Council of Canada | B Jesse Shapiro |
| Canada Research Chairs | B Jesse Shapiro |

The funders had no role in study design, data collection and interpretation, or the decision to submit the work for publication.

### Author contributions

Naïma Madi, Conceptualization, Data curation, Software, Formal analysis, Investigation, Visualization, Methodology, Writing - original draft, Writing - review and editing; Michiel Vos, Conceptualization, Data curation, Supervision, Investigation, Writing - original draft, Writing - review and editing; Carmen Lia Murall, Software, Formal analysis, Validation, Investigation, Methodology, Writing - review and editing; Pierre Legendre, Conceptualization, Methodology, Writing - review and editing; B Jesse Shapiro, Conceptualization, Formal analysis, Supervision, Funding acquisition, Investigation, Methodology, Writing - original draft, Project administration, Writing - review and editing

### Author ORCIDs

Carmen Lia Murall  http://orcid.org/0000-0002-1543-4501
B Jesse Shapiro  https://orcid.org/0000-0001-6819-8699

### Decision letter and Author response

Decision letter https://doi.org/10.7554/eLife.58999.sa1
Author response https://doi.org/10.7554/eLife.58999.sa2

## Additional files

### Supplementary files

- Supplementary file 1. Full GLMM outputs for the EMP data.
- Supplementary file 2. Goodness of fit for the GLMMs.
- Supplementary file 3. Full GLMM output for simulated data under Neutral Theory models.

• Supplementary file 4. Full GLMM output for soil data (*Delgado-Baquerizo et al., 2018*).

• Supplementary file 5. Indicator species analysis. The table shows the assignment of each genus to one of three environment types.

• Supplementary file 6. Genome size assignment. The table shows genome sizes assigned to each genus.

• Transparent reporting form

## Data availability

All data is available from the Earth Microbiome Project (ftp.microbio.me), as detailed in the Methods. All computer code used for analysis are available at https://github.com/Naima16/dbd.git (copy archived at https://archive.softwareheritage.org/swh:1:rev:ecb4f844264b72eaa8cb-d708244ecd32d414c7dd/).

The following previously published datasets were used:

| Author(s) | Year | Dataset title | Dataset URL | Database and Identifier |
|---|---|---|---|---|
| Thompson LR, Sanders JG, McDonald D, Amir A, Ladau J, Locey K, Prill RJ, Tripathi A, Gibbons SM, Ackermann G, Navas-Molina JA, Janssen S, Kopylova E, Vázquez-Baeza Y, González A, Morton JT, Mirarab S, Xu ZZ, Jiang L, Haroon MF, Kanbar J, Zhu Q, Song SJ, Kosciolek T, Bokulich NA, Lefler J, Brislawn CJ, Humphrey G, Owens SM, Hampton-Marcell J, Berg-Lyons D, McKenzie V, Fierer N, Fuhrman JA, Clauset A, Stevens RL, Shade A, Pollard RS, Goodwin KD, Jansson JK, Gilbert JA, Knight R | 2017 | Earth Microbiome Project | ftp://ftp.microbio.me/emp/release1/otu_distributions/otu_summary.emp_deblur_90bp.subset_2k.rare_5000.tsv | microbio.me, otu_summary.emp_deblur_90bp.subset_2k.rare_5000.tsv |
| Delgado-Baquerizo M, Oliverio AM, Brewer TE, Benavent-González A, Eldridge DJ, Bardgett RD, Maestre FT, Singh BK, Fierer N | 2018 | Global Soil Dataset | https://figshare.com/s/82a2d3f5d38ace925492 | figshare, 10.6084/m9.figshare.5611321.v3 |

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
