## [Decision Letter]

**Acceptance summary:**

This study revisits an old debate in the ecology literature dating back to the 1940s: Does diversity promote or constrain diversity at different levels, and how is diversity maintained? These questions have been investigated in communities of macroorganisms, but this is probably the first high-level analysis of a large microbiome dataset to explore how species interactions influence microbial diversity. The authors found support for the "Diversity Begets Diversity" (DBD) hypothesis in habitats with lower overall diversity, while in habitats with higher overall diversity, the "Ecological Controls" (EC) hypothesis appeared to more closely describe patterns of microbial diversity. The paper has a great mix of modern tools, historical ideas, and interdisciplinary science. It is clearly written and the analyses are generally carefully executed and presented.

**Decision letter after peer review:**

Thank you for submitting your article "Does diversity beget diversity in microbiomes?" for consideration by *eLife*. Your article has been reviewed by three peer reviewers, and the evaluation has been overseen by Detlef Weigel as the Reviewing and Senior Editor. The following individuals involved in review of your submission have agreed to reveal their identity: Eric Kemen (Reviewer #1); Benjamin E Wolfe (Reviewer #2).

The reviewers have discussed the reviews with one another and the Reviewing Editor has drafted this decision to help you prepare a revised submission.

Summary:

Madi and colleagues revisit an old debate in the ecology literature dating back to the 1940s: Does diversity promote or constrain diversity at different levels, and how is diversity maintained? On one hand, having more species to interact with can create new niches and promote diversity. On the other hand, more species might mean more competition that would limit additional gains in diversity. While these hypotheses make opposite predictions, they are not mutually exclusive and likely act in tandem during community assembly. As the authors note, these questions have been asked in communities of macroorganisms, but this is probably the first high-level analysis of a large microbiome dataset that explores how species interactions influence microbial diversity.

The authors found support for the "Diversity Begets Diversity" (DBD) hypothesis in habitats with lower overall diversity, while in habitats with higher overall diversity, the "Ecological Controls" (EC) hypothesis appeared to more closely describe patterns of microbial diversity.

There are many caveats when inferring microbial interactions and other dimensions of microbial diversity from 16S rRNA amplicon data, but this work provides a great jumping point for many future studies to experimentally test some of the conclusions of this analysis.

The paper has a great mix of modern tools, historical ideas, and interdisciplinary science. It is clearly written and the analyses are generally carefully executed and presented.

However, there are also several significant concerns, which we hope the authors will be able to address.

Essential revisions:

1) A third significant hypothesis is missing that makes predictions about how diversity can affect net rates of diversification, The Neutral Theory of Biodiversity and Biogeography (NT). The pattern the authors observed – decreased diversity slopes within clades embedded within more diverse communities – can be explained by a neutral model devoid of ecological interactions. Let us consider a simple neutral model that has a rate of diversification intrinsic to each unique taxonomic unit (new taxa emerging per generation per unique taxonomic unit) and a rate of stochastic extinction that is proportional to the population size of each taxonomic unit. The probability that any taxa diversifies would be uniform throughout the community, however, as the diversity increases, this increases the number of unique units that can yield new taxa and thus the rate of influx of new taxa. At some tipping point, however, the community will become so diverse that the population size of any given taxa will be low and susceptible to stochastic loss. This will cause the net diversification rate to drop in more diverse communities. This is just a simple example and it does not consider migration, which is important for shaping microbial diversity; however, it demonstrates that other forces outside of the ecologically-based models of DBD and EC may be acting to produce the patterns under investigation. It is important to rule out NT's parsimonious predictions before moving on to testing sophisticated ecological theories.

2) Following from the NT discussion, there is likely an important role for extinction in shaping these patterns. The discussion of hypotheses focused on the production of diversity and not the net balance of production and loss. This omission is a major weakness of the paper; some discussion of the role of extinction in these patterns would strengthen the interpretations.

3) The analysis of the relationship between genome size and DBD in the context of the Black Queen Hypothesis is not convincing. As the authors acknowledge, there are serious limitations with assuming that the genome size from one strain/species within a genus can fully represent the genome size of that genus. The authors do not provide an estimate of how much error might result from this approach (for example, by looking at several genera where many genome sizes exist and seeing how that impacts their analyses). They also do not fully explain their rationale for some aspects of this analysis; for example, why was the largest genome size available used? Why would having more precise genome data make the positive relationship stronger?

We suggest that this part of the analysis is removed in the revision. You may refer to it in the Discussion as speculation.

If you decide to discuss genome size and how this could correlate with more metabolic functions, you should elaborate further on this. Generally genome size correlates to repeats rather than additional metabolite clusters. You could use the available genomes to annotate genome clusters (using e.g. antiSMASH).

4) One of the major findings is the evidence for DBD being strongest in less diverse biomes and weaker in more diverse biomes. However, some of these biomes also suggest a correlation with total microbial load. For example, the animal distal gut is a less diverse biome but an extremely densely populated biome, whereas the soil biome is exceptionally diverse but has a lower total microbial density than the distal animal gut. Do the authors have access to any data about the population density in these different biomes? Does population density also correlate with the DBD-EC continuum? If the authors do not think that total microbial load factors into the DBD-EC continuum, they should provide an explicit rationale why not. Note that we do not expect that microbial population size data are available for the entire data set, but perhaps there is a subset for which this is available and that could be analysed.

5) For model validation for GLMMs, it would be useful if the authors could report goodness of fit for all their models (differences between the observed values and the model's predicted) using the Akaike's Information Criterion (AIC) or the Bayesian Information Criterion (BIC). Further, using cross validation (dividing data to train and test) might be useful to estimate the error rate of the models and could be used to confirm the accuracy of the models.

Additional major comments:

6) The measures for diversity vary over the paper and it is not always clear when which measurement is used, which makes it difficult to compare the analyses. Where the measures differ, this should be justified.

For example, this is clear in Figures 1, 4 and 5, but not in Figure 2. Similarly, Figure 2—figure supplement 1, which show the number of focal taxa as a function of the number of non-focal taxa, but in Figure 3 the diversity slope was estimated from a GLMM using taxonomic ratio for the community diversity, without mentioning what diversity measure was used for focal lineage. More generally, are taxonomic ratios a good measures of lineage diversity? Please explain why you used these measures of diversity instead of others. You mention that you also tried others such as Shannon index that are also robust but did not follow up on this.

7) In the very beginning of the Results, the authors note that a null model was used to assess the slopes from their GLMMs (subsection “Quantifying the DBD-EC continuum in prokaryote communities”), but details are provided only later in the Materials and methods. We think the paper would be very much improved if the authors spent a paragraph explaining the null model early in the Results section and if a data figure (or figures) from the null model were moved to the main body of the text from the supplementary information. This would help the reader understand how support for significance of the slopes was determined as they move through the rest of the paper. More importantly, the paper would have greater readability and impact for a broader audience if the authors explained the development and rationale of their null model in very simple terms.

8) The term ASV is used throughout, but the downloaded databases contain mainly OTUs. This of course could affect the outcome of an analyses depending if OTUs and ASVs are mixed or either OTUs or ASVs are used. Please explain in more detail what was used and how it was calculated.

9) A major limitation that is not acknowledged is a sole focus on prokaryotic taxa. Many of the ecosystems sampled in the EMP dataset have diverse and abundant fungi, protists, and other types of microbes. It is likely that these other microbial taxa interact with the target bacteria studied in this work in diverse ways (as numerous previous studies have shown). The authors should acknowledge this major limitation and explore briefly how it may impact their findings in the Discussion of the text. For example, fungi may play disproportionate roles in some environments that explain some of the variation observed here (e.g. the rhizosphere).

10) Subsection “Abiotic drivers of diversity”, last two paragraphs: These two paragraphs contain two analyses that essentially contradict each other. These varying results are interesting – could you please expand a little more on why these two analyses might be showing different results? In particular, the last sentence suggests that diversity levels in soil and other communities with a DBD plateau are predominantly controlled by abiotic factors. However, this is the first mention of those specific biomes in this paragraph. Could you add a little more about this observation?

---

## [Author Response]

Essential revisions:1) A third significant hypothesis is missing that makes predictions about how diversity can affect net rates of diversification, The Neutral Theory of Biodiversity and Biogeography (NT). The pattern the authors observed – decreased diversity slopes within clades embedded within more diverse communities – can be explained by a neutral model devoid of ecological interactions. Let us consider a simple neutral model that has a rate of diversification intrinsic to each unique taxonomic unit (new taxa emerging per generation per unique taxonomic unit) and a rate of stochastic extinction that is proportional to the population size of each taxonomic unit. The probability that any taxa diversifies would be uniform throughout the community, however, as the diversity increases, this increases the number of unique units that can yield new taxa and thus the rate of influx of new taxa. At some tipping point, however, the community will become so diverse that the population size of any given taxa will be low and susceptible to stochastic loss. This will cause the net diversification rate to drop in more diverse communities. This is just a simple example and it does not consider migration, which is important for shaping microbial diversity; however, it demonstrates that other forces outside of the ecologically-based models of DBD and EC may be acting to produce the patterns under investigation. It is important to rule out NT's parsimonious predictions before moving on to testing sophisticated ecological theories.

We thank the reviewers for pointing out this oversight on our part; the Neutral Theory (NT) is a natural concept to include, and we now consider it explicitly as a potential explanation for the diversity slopes observed in the EMP data. We mention the NT in the Introduction, and the Results now include simulated data sampled from the distribution that arises from the classical NT: the zero-sum multinomial (ZSM). These simulated data are generated in a very similar way to our previous null models, except that the Poisson distribution has been replaced with the ZSM to align with NT. As before, and as suggested by the reviewers, ours is a model of community assembly rather than de novo diversification, which is more appropriate for the 16S amplicon data from EMP (in which de novo speciation within a sample is highly unexpected).

Under neutral Model 1, in which all data are sampled from the same ZSM distribution, there is a significantly negative diversity slope, and the trend is visibly linear (updated Figure 2—figure supplement 7). Under neutral Model 2, in which each environment draws from a separate distribution (which we consider to be realistic given that environments have systematically different levels of community diversity), there is a visibly positive trend (due to ‘stacked’ negative slopes from different environments; Figure 2—figure supplement 7) but this is found to be non-significant in a GLMM that accounts for the effect of different environments. We note that our previous Model 2 (which was made with a Poisson distribution) contained an error: although we stated in the text that there was one distribution per environment, in fact the data were generated from a model with one distribution per sample. This error has now been corrected, and we consider the ‘environment-specific diversity’ model more parsimonious than the ‘sample-specific diversity’ model, especially given that sample-specific effects on the diversity slope are relatively small (Table 1). The neutral null models 1 and 2 assume no species interactions by definition and so are not included as ‘baselines’ for the DBD or EC “spike-in” simulations. We keep the old Model 1 (sampled from one Poisson distribution) as the baseline for the DBD and EC “spike-in” simulations, and it is now called Model 3.

Overall, the updated model results suggest that a positive diversity slope is unlikely to arise under NT, but that negative slopes can arise. Combined with the finding that adding DBD, but not EC, is detectable by our GLMMs, this suggests we are well-powered to detect DBD but likely underpowered to detect EC in the EMP data. Therefore, the positive slopes in the EMP data are likely real, but there may be undetectable EC effects as well.

Regarding the plateau of DBD at high levels of diversity: these results indicate that the ‘upward’ part of the slope is likely due to DBD, but the plateau could be due to a combination of EC and sampling from a NT model. We also note that the observed ‘plateau’ of DBD in more diverse biomes (shown in Figure 3) level is not observed under the neutral Model 2. Therefore, this DBD plateau cannot be easily explained by NT.

To summarize, we are able to rule out NT’s parsimonious predictions in the sense that they are unlikely to give rise to the positive DBD slopes observed. However, it is hard to tell whether negative slopes in the EMP data arise from NT, EC, or a mixture of the two. This is in line with previous studies that have found distinguishing competition and neutrality based on metacommunity data to be non-trivial. Further work and more sophisticated models will be needed to tease these apart.

2) Following from the NT discussion, there is likely an important role for extinction in shaping these patterns. The discussion of hypotheses focused on the production of diversity and not the net balance of production and loss. This omission is a major weakness of the paper; some discussion of the role of extinction in these patterns would strengthen the interpretations.

In the absence of a substantial fossil record, it is very difficult to accurately deconvolute speciation and extinction rates from phylogenies and instead we must rely on the net diversification rate (speciation minus extinction) (Louca and Pennell, 2020; Marshall, 2017). Nevertheless, Louca et al. recently estimated that bacterial diversity has mostly increased over the past billion years, with speciation rates slightly exceeding extinction rates (Louca et al., 2018). (We note that this is generally consistent with the widespread action of DBD, and less consistent with widespread EC). Thus, although bacterial lineages do go extinct, the precise rates of extinction and speciation are not straightforward to infer from phylogenies, and we do not attempt this here. More importantly, we do not think that speciation and extinction dynamics are at play at the scale of 16S rRNA gene sequencing in the EMP dataset. Although molecular clocks vary across bacterial lineages, the 16S gene typically evolves at a substitution rate of <0.1% per million years, equivalent to 1-2 substitutions (Kuo and Ochman, 2009b). Therefore, even at the resolution of ASVs, we do not expect new ASVs to evolve within EMP samples (and this is even more true for higher taxa like genera or phyla). We thus consider most of our observations to be consistent with an ‘ecological community assembly’ rather than ‘evolutionary diversification’ model of DBD. These points have now been added to the Introduction.

3) The analysis of the relationship between genome size and DBD in the context of the Black Queen Hypothesis is not convincing. As the authors acknowledge, there are serious limitations with assuming that the genome size from one strain/species within a genus can fully represent the genome size of that genus. The authors do not provide an estimate of how much error might result from this approach (for example, by looking at several genera where many genome sizes exist and seeing how that impacts their analyses). They also do not fully explain their rationale for some aspects of this analysis; for example, why was the largest genome size available used? Why would having more precise genome data make the positive relationship stronger?We suggest that this part of the analysis is removed in the revision. You may refer to it in the Discussion as speculation.If you decide to discuss genome size and how this could correlate with more metabolic functions, you should elaborate further on this. Generally genome size correlates to repeats rather than additional metabolite clusters. You could use the available genomes to annotate genome clusters (using e.g. antiSMASH).

We agree that this analysis was speculative and it has now been clearly flagged as such and moved to the Discussion. Although this is a preliminary analysis, the effects of biotic interactions on genome size evolution (including the Black Queen Hypothesis) is a subject of growing interest, and we believe our analysis will spur discussion and further study. The Black Queen Hypothesis is now only briefly mentioned in the Discussion. We plan to follow up on these genome size analyses using higher-resolution genomic or metagenomic data, and hopefully publish these as a Research Advance as suggested above by the editor (https://elifesciences.org/articles/57162). In the meantime, we attempt to clarify and justify the exploratory genome size analyses as much as possible.

First, we justify the choice of using the largest genome size in a genus as in the Materials and methods section as follows: “If multiple genomes were available with the same level of completion, the largest genome size was used, as smaller genomes could be artefacts of incomplete assembly which would bias the mean and median downward. Moreover, given the deletional bias in bacterial genomes (Kuo and Ochman, 2009a), the largest genome is likely more reflective of the ancestral genome size of the genus.”

Second, we further explain why having more precise genome size data would make the positive relationship stronger, in the Discussion as follows: “The positive relationship between genome size and DBD is likely even stronger than estimated, because assigning genome sizes to entire genera is imprecise (i.e. there is variation in genome size within a genus, or even within species), therefore weakening the correlation.”

In essence, we argue that we see a DBD-genome size relationship *in spite of* a noisy estimate of genome size. If the relationship is real, having less noisy estimates would improve the correlation.

Finally, given that estimating genome size at the genus level based on 16S taxonomic information is already noisy, we refrain from running additional analyses (e.g. antiSMASH) based on this dataset, and prefer to follow up on more appropriate genomic datasets. Nevertheless, we believe it is a reasonable assumption that the number of gene families (and thus metabolic potential, in a general sense) scale with genome size in bacteria. In bacteria and archaea, there is a linear (log-log) correlation between genome size and the number of protein-coding genes, and the linear relationship reaches a plateau in eukaryotes

(https://commons.wikimedia.org/wiki/File:Genome_size_vs_protein_count.svg). This is likely because larger eukaryotic genomes contain more repeats, transposons, and viral sequences, but this is not the case for bacteria, as shown for example in a study of soil bacteria in which genome size is correlated with the number of coding genes, including secondary metabolic pathways (Barberán et al., 2014). We also note that the range of bacterial genome size variation (1-15 Mbp) that we find affect diversity slope (in our Figure 5) is too great to be explained simply by repeat sequences, and must be largely driven by coding sequences. Therefore, the literature clearly supports a scaling of bacterial genome size with protein coding genes, and likely with metabolic potential. Nevertheless, the relevance of this to DBD remains speculative and deserves further study, as clearly stated in our revised Discussion section.

4) One of the major findings is the evidence for DBD being strongest in less diverse biomes and weaker in more diverse biomes. However, some of these biomes also suggest a correlation with total microbial load. For example, the animal distal gut is a less diverse biome but an extremely densely populated biome, whereas the soil biome is exceptionally diverse but has a lower total microbial density than the distal animal gut. Do the authors have access to any data about the population density in these different biomes? Does population density also correlate with the DBD-EC continuum? If the authors do not think that total microbial load factors into the DBD-EC continuum, they should provide an explicit rationale why not. Note that we do not expect that microbial population size data are available for the entire data set, but perhaps there is a subset for which this is available and that could be analysed.

We thank the reviewers for raising this excellent point. As the reviewers suspected, we do not have comprehensive estimates of microbial population sizes (cell counts) for the entire dataset, but a subset allows us to make some informed speculation. Indeed, the animal distal gut has a density ~10^11^ cells/mm3^3^ and has relatively low diversity while soil is more diverse but less dense (~107 -109 cells/mm ). However, cell density does not appear to correlate with the DBD-EC continuum, as is now illustrated in the updated Figure 3. For example, saline water has an intermediate level of diversity compared to the distal gut or soil, but has lower cell density than either of these biomes (~10^6^ cells/mm^3^ ). Non-saline water has a similar density, but tends to have higher diversity than saline water. Of course, these are rough estimates and a more systematic study could allow a more refined analysis, but we tentatively conclude that differential cell density across biomes is unlikely to explain variation along the DBD-EC continuum. We suspect this is because even relatively low-density microbiomes yield sufficient numbers of 16S amplicons for analysis. These findings are now reported in the Results describing Figure 3.

5) For model validation for GLMMs, it would be useful if the authors could report goodness of fit for all their models (differences between the observed values and the model's predicted) using the Akaike's Information Criterion (AIC) or the Bayesian Information Criterion (BIC). Further, using cross validation (dividing data to train and test) might be useful to estimate the error rate of the models and could be used to confirm the accuracy of the models.

As suggested, we have calculated AIC for full GLMM models, and compared these to models without any fixed effects save the intercept. The delta-AIC values for these comparisons, as well as likelihood-ratio test *p*-values are now reported in the new Supplementary file 1. As an additional measure of goodness of fit, we also report both marginal (including only fixed effects) and conditional (including both fixed and random effects) coefficients of variation (*R^2^*) in Supplementary file 1. In general, the conditional *R^2^* values are higher, suggesting the importance of random effects. This is now mentioned briefly in the first Results paragraph, and explained in detail in the Materials and methods.

Additional major comments:6) The measures for diversity vary over the paper and it is not always clear when which measurement is used, which makes it difficult to compare the analyses. Where the measures differ, this should be justified.For example, this is clear in Figures 1, 4 and 5, but not in Figure 2. Similarly, Figure 2—figure supplement 1, which show the number of focal taxa as a function of the number of non-focal taxa, but in Figure 3 the diversity slope was estimated from a GLMM using taxonomic ratio for the community diversity, without mentioning what diversity measure was used for focal lineage. More generally, are taxonomic ratios a good measures of lineage diversity? Please explain why you used these measures of diversity instead of others. You mention that you also tried others such as Shannon index that are also robust but did not follow up on this.

We apologize for this confusion, and we have now clarified the metrics of focal lineage diversity and community diversity used throughout the manuscript. Briefly, focal lineage diversity is always measured as a taxonomic ratio in all the main text figures, and the axis labels have now been adjusted to clarify this. The only alternative metric of focal lineage diversity is when we use percent nucleotide identity instead of taxonomic ratios, as in Figure 2—figure supplements 10 and 11. In the main text figures, community diversity is always measured as the number of non-focal taxa. The only exception is a sensitivity analysis, in which we show that using Shannon diversity as an alternative metric of community diversity (Table 3) gives qualitatively similar results as using the number of non-focal taxa as in other analyses (summarized in Table 1). We have made clarifications throughout the manuscript, notably in the Figure 2 legend to specify that these are taxonomic ratios as in Figure 1.

We chose to focus the primary analyses on taxonomic ratios because they are simple, interpretable (i.e. they allow linking the results to well-studied organisms, e.g*. Pseudomonas*), and have a long history in the ecological literature dating back to Elton. We view the taxonomy-independent analyses (using percent nucleotide identity in the 16S rRNA gene) as supportive of the primary results based on taxonomic ratios, showing that these primary results are not overly biased by taxonomy. Similarly, we use Shannon diversity as an alternative measure of community diversity (which is otherwise measured simply as richness, or the number of non-focal taxa) to show that major findings are not sensitive to the choice of metric. We consider these controls using alternative diversity metrics to be sufficient for our broad-scale analyses, and leave it for future investigations to investigate details of why different metrics might differ slightly, tracking different aspects of the EC-DBD continuum. For the time being, we hope it is now clear which measures were used for which analyses in our manuscript, and for what reason.

7) In the very beginning of the Results, the authors note that a null model was used to assess the slopes from their GLMMs (subsection “Quantifying the DBD-EC continuum in prokaryote communities”), but details are provided only later in the Materials and methods. We think the paper would be very much improved if the authors spent a paragraph explaining the null model early in the Results section and if a data figure (or figures) from the null model were moved to the main body of the text from the supplementary information. This would help the reader understand how support for significance of the slopes was determined as they move through the rest of the paper. More importantly, the paper would have greater readability and impact for a broader audience if the authors explained the development and rationale of their null model in very simple terms.

We fully agree, and a description of the null models (now explicitly linked to Neutral Theory, as described in the response to Essential Revision #1 above) now appears prominently in the first two Results paragraphs.

8) The term ASV is used throughout, but the downloaded databases contain mainly OTUs. This of course could affect the outcome of an analyses depending if OTUs and ASVs are mixed or either OTUs or ASVs are used. Please explain in more detail what was used and how it was calculated.

We aimed to make all our analyses as standardized and comparable as possible, we used ASVs throughout – for both EMP and global soil datasets. To make the soil dataset comparable to the EMP, we downloaded the raw sequencing data and called ASVs using deblur, using the same pipeline as for the EMP. This is now clarified in the Results paragraph introducing the soil dataset analysis.

9) A major limitation that is not acknowledged is a sole focus on prokaryotic taxa. Many of the ecosystems sampled in the EMP dataset have diverse and abundant fungi, protists, and other types of microbes. It is likely that these other microbial taxa interact with the target bacteria studied in this work in diverse ways (as numerous previous studies have shown). The authors should acknowledge this major limitation and explore briefly how it may impact their findings in the Discussion of the text. For example, fungi may play disproportionate roles in some environments that explain some of the variation observed here (e.g. the rhizosphere).

We appreciate this suggestion, and fully agree. We now mention in the Discussion how interactions involving non-prokaryotes such as phages and eukaryotes could explain some of the variation along the DBD-EC continuum.

10) Subsection “Abiotic drivers of diversity”, last two paragraphs: These two paragraphs contain two analyses that essentially contradict each other. These varying results are interesting – could you please expand a little more on why these two analyses might be showing different results? In particular, the last sentence suggests that diversity levels in soil and other communities with a DBD plateau are predominantly controlled by abiotic factors. However, this is the first mention of those specific biomes in this paragraph. Could you add a little more about this observation?

We have changed the end of this paragraph to further reconcile the apparent contradiction between EMP and soil datasets. The result that DBD is weak and abiotic drivers of diversity are strong in soil can be reconciled with the finding of generally stronger DBD in the overall EMP dataset. We showed that soil has a relatively low diversity slope, even without considering abiotic factors (Figure 3). It is therefore not surprising to estimate a low diversity slope when abiotic factors are added to the model. This is now explained on in the subsection “Abiotic drivers of diversity”.